# Drug screening to identify compounds to act as co-therapies for the treatment of *Burkholderia* species

Sam Barker[1¤], Sarah V. Harding[2], David Gray[3], Mark I. Richards[2], Helen S. Atkins[1,2], Nicholas J. Harmer[1,4]*

**1** Department of Biosciences, College of Life and Environmental Sciences, University of Exeter, Exeter, United Kingdom, **2** Defence Science and Technology Laboratory, Porton Down, Salisbury, United Kingdom, **3** Drug Discovery Unit, University of Dundee, Dundee, Scotland, **4** Living Systems Institute, Exeter, United Kingdom

¤ Current address: Midatech Pharma Plc, Cardiff, United Kingdom
* N.J.Harmer@exeter.ac.uk

**Data Availability Statement:** Data underpinning this work are available through Open Research Exeter at https://doi.org/10.24378/exe.2963.

## Abstract

*Burkholderia pseudomallei* is a soil-dwelling organism present throughout the tropics. It is the causative agent of melioidosis, a disease that is believed to kill 89,000 people per year. It is naturally resistant to many antibiotics, requiring at least two weeks of intravenous treatment with ceftazidime, imipenem or meropenem followed by 6 months of orally delivered co-trimoxazole. This places a large treatment burden on the predominantly middle-income nations where the majority of disease occurs. We have established a high-throughput assay for compounds that could be used as a co-therapy to potentiate the effect of ceftazidime, using the related non-pathogenic bacterium *Burkholderia thailandensis* as a surrogate. Optimization of the assay gave a Z' factor of 0.68. We screened a library of 61,250 compounds and identified 29 compounds with a $pIC_{50}$ ($-\log_{10}(IC_{50})$) greater than five. Detailed investigation allowed us to down select to six "best in class" compounds, which included the licensed drug chloroxine. Co-treatment of *B. thailandensis* with ceftazidime and chloroxine reduced culturable cell numbers by two orders of magnitude over 48 hours, compared to treatment with ceftazidime alone. Hit expansion around chloroxine was performed using commercially available compounds. Minor modifications to the structure abolished activity, suggesting that chloroxine likely acts against a specific target. Finally, an initial study demonstrates the utility of chloroxine to act as a co-therapy to potentiate the effect of ceftazidime against *B. pseudomallei*. This approach successfully identified potential co-therapies for a recalcitrant Gram-negative bacterial species. Our assay could be used more widely to aid in chemotherapy to treat infections caused by these bacteria.

## Introduction

*Burkholderia pseudomallei* is the causative agent of melioidosis, a disease endemic to many regions across the tropics [1]. It is believed to cause approximately 89,000 deaths per annum

**Funding:** The authors note that author SB was employed by Phoremost Ltd and Midatech Plc after completing his work on this manuscript, but before the final manuscript was completed. These companies provided his salary but did not have any influence on the text or data of the manuscript, or the decision to publish. This does not alter our adherence to PLOS ONE policies on sharing data and materials. The authors declare no other competing interests. This work was funded by grant Dstlx-1000060221 from Dstl to NJH.

**Competing interests:** The authors note that author SB was employed by Phoremost Ltd and Midatech Plc after completing his work on this manuscript, but before the final manuscript was completed. These companies provided his salary but did not have any influence on the text or data of the manuscript, or the decision to publish. This does not alter our adherence to PLOS ONE policies on sharing data and materials. The authors declare no other competing interests.

worldwide [2,3], with the large majority of the burden falling on less developed or lower middle income countries. Melioidosis can present in many ways, which significantly complicates diagnosis [4]. Clinical presentations include skin infections, suppurative parotitis, genitourinary infections, and pneumonia [5]. The most serious infections can develop to sepsis, and abscesses on internal organs are common [1,6]. In the absence of treatment, mortality from acute infections is high; even with treatment, mortality approaches 40% in many affected areas [7]. Patients with access to adequate diagnostic and treatment facilities have reduced mortality rates [8] and are treated with an intensive treatment phase of intravenously delivered ceftazidime, imipenem or meropenem for at least 14 days [9–11], followed by oral eradication therapy with co-trimoxazole lasting between 3 and 6 months [1,12,13]. The cost of this treatment regime is high and the burden of disease in the least developed countries (e.g. Cambodia) may prevent those in need from being treated [14,15]. In many lower income settings alternative eradication regimes are used that have increased disease relapse rates [16].

*B. pseudomallei* is found in soil and water, preferring anthrosol and acrisol soil types [2,3]. Like many *Burkholderia*, it is an opportunistic pathogen of humans, and most patients have at least one pre-disposing risk factor (with diabetes mellitus the most common) [17]. In the host, *B. pseudomallei* generally adopts an intracellular lifestyle, and can invade and replicate in a range of cell types [18]. The intracellular location also makes antibiotic chemotherapy more challenging as compounds must cross an additional biological membrane.

*B. pseudomallei* is naturally resistant to many clinically used antibiotics, including some of the more recently developed antibiotics [1,9,10,19]. When cultured to stationary phase or in hypoxic conditions, most *Burkholderia* species show a high subpopulation that are recalcitrant to antibiotic treatment [20]. This observation is believed to mimic behavior *in vivo*, with *B. pseudomallei* surviving in biofilms or intracellular niches where cellular conditions promote antibiotic tolerance [21–23]. This can then lead to recurrent or latent forms of the disease and the relapse of infections in humans where longer term antibiotic treatment is not administered [24]. Although significant progress has been made towards a melioidosis vaccine, candidates are yet to enter clinical trials [25,26].

This presents an urgent unmet need for affordable novel drugs that supplement current effective therapeutics to reduce the cost and duration of treatment and to prevent relapse of infection [11,27]. We hypothesized that small molecules could act as co-therapies that could be administered alongside front-line treatments with the aim of reducing the rates of recurrent infection. We aimed to develop an assay that would allow rapid screening of a compound library to identify and validate such compounds, as a step towards a potential therapy. As *B. pseudomallei* is a Containment Level 3 bacterium, *Burkholderia thailandensis* was selected for this study. This is a close relative of the pathogenic *B. pseudomallei* with over 85% gene conservation [28]. As *B. thailandensis* does not cause disease in immunocompetent humans [28–30], it is commonly used as a surrogate for *B. pseudomallei*. Previous studies have shown that approximately 0.1% of *B. thailandensis* cells survive for 24 hours following treatment with 100X MIC (minimum inhibitory concentration; the lowest concentration of an antibiotic required to prevent observable growth of the bacterium; c.f. 5–10% of *B. pseudomallei* survive such treatment) of the front-line antibiotic ceftazidime *in vitro* [20].

A phenotypic assay using the cell viability reagent PrestoBlue™ was used to screen compounds from a diversity library containing nearly 5,000 core fragments [31] at the Drug Discovery Unit (DDU) in Dundee. Preliminary screening identified six compounds that were active as co-therapies and potentiated the effect of ceftazidime against *B. thailandensis*. Following hit confirmation and potency determination, we identified chloroxine, which had an $IC_{50}$ (concentration at which 50% of the maximal growth inhibition is observed) value lower than 10 µM, as the most promising compound. Chloroxine was able to reduce the proportion of

cells surviving ceftazidime treatment by at least two orders of magnitude. Evaluation of structurally similar compounds suggested that chloroxine has a specific effect. This study suggests that chloroxine has strong potential for further development as a ceftazidime co-therapy for melioidosis.

## Materials and methods

### Bacterial strain and culture conditions

*B. thailandensis* strain E264 (ATCC; strain 700388) was grown in high salt (10 g/L) Lysogeny broth (LB) at 37˚C with aeration at 200 rpm. For experiments investigating the activity of the combination therapy, *B. thailandensis* was grown to stationary phase in LB broth and cells harvested by centrifugation. Cell pellets were resuspended in M9 minimal media [32] supplemented with 730 μM/400 μg/ml ceftazidime hydrate (Melford Laboratories, #C5920; hereafter referred to as ceftazidime). Initial cell counts were determined from the absorbance at 600 nm. An $OD_{600}$ of 0.2 corresponds to $2x10^8$ cfu (Claudia Hemsley, University of Exeter, personal communication). For growth of *B. pseudomallei* strain K96243 (S. Songsivilai, Siriraj Hospital), bacteria were plated onto low salt (5 g/L) LB-agar. Single colonies were picked into 100 ml low salt LB broth and grown at 37˚C for 20 hours with orbital shaking. Cells were harvested by centrifugation and pellets resuspended in M9 minimal media. Ceftazidime was prepared from a stock at 73 mM active component in 0.1 M sodium hydroxide. Chloroxine (Sigma-Aldrich, #D64600) was prepared from a stock at 10–100 mg/ml active component in dimethyl sulfoxide (DMSO).

### Cell viability assay

Detection of cell viability with PrestoBlue™ (Life Technologies, #A13261) was performed in 96 and 384 well, black walled assay plates (Corning, #3904 and #3573 respectively) by adding 10% PrestoBlue (v/v) to each bacterial culture. Following the addition of PrestoBlue, plates were incubated at room temperature for one hour and fluorescence was read at ex 540/em 590 nm by an Envision plate reader (PerkinElmer), or an Infinite M200 Pro (Tecan). All liquid handling in the primary screen and hit expansion was automated.

An assay was developed to discriminate two-fold changes in cell numbers. A bacterial culture was prepared as described above and serially diluted in an equal volume of M9 media to produce two-fold dilutions. A positive control (cells resuspended in M9 media without ceftazidime) and a negative control (cells heat killed at 90˚C for 2 minutes) were included in these assays. Plates were incubated at 37˚C overnight before addition of PrestoBlue reagent and the reading of fluorescence as described previously.

### High throughput screening

A library of 61,250 compounds was prepared as stock solutions in DMSO at a concentration of 10 mM and supplied in 384-well Echo plates (Labcyte, #P-05525) for use in this screen. 45 μl of a culture resuspended in M9 media supplemented with 730 μM ceftazidime to an $OD_{600\ nm}$ of 0.8 (equivalent to late log phase growth, equivalent to $8x10^8$ cfu/mL) was added to give a final compound concentration of 30 μM. Plates were covered with AeraSeal film (Sigma-Aldrich, #A9224) before incubation for 24 hours at 28˚C. A single point (SP) screen of all compounds was performed.

309 compounds from the diversity library were tested for potency using a standard ten point half logarithm concentration response protocol [33]. Selected hits were dispensed into 384 well Echo plates using a Biomek FX automated liquid handling workstation (Beckman

Coulter); two-fold serial dilutions of each compound in DMSO was performed using an Echo 550 liquid handler (Labcyte).

Our specifications for assay design stipulated a Z factor > 0.5 [34,35].

$$Z\ Factor = 1 - \frac{3(\sigma p + \sigma n)}{|\mu p - \mu n|} \tag{1}$$

Where $\mu p$ and $\sigma p$ are the mean and standard deviation of cells treated with ceftazidime, and $\mu n$ and $\sigma n$ are the mean and standard deviation of the negative controls. A worked example calculation is available in the legend to S1 Fig.

## Data processing and analysis

Data analysis was performed within ActivityBase (IDBS) and report creation was undertaken using Vortex (Dotmatics). All $IC_{50}$ curve fitting was undertaken within Activity Base XE utilizing the underlying 'MATH IQ' engine of XLfit version 5.1.0.0 from IDBS. Curve fitting was carried out using the following 4 parameter logistic equation:

$$y = A + \frac{(B - A)}{1 + \left(\frac{10^C}{x}\right)^D} \tag{2}$$

where A = % inhibition at bottom, B = % inhibition at top, C = 50% effect concentration ($IC_{50}$), D = slope, $x$ = inhibitor concentration and y = % inhibition. As $IC_{50}$ values are Log normally distributed, fitted $IC_{50}$ values are stated as the $pIC_{50}$ ($-log_{10}[IC_{50}]$).

## Minimum inhibitory concentration (MIC)

These were determined following the CLSI recommended protocol for antimicrobial susceptibility testing via micro dilution method [36,37]. Experiments were initiated with an inoculum of approximately $1 \times 10^5$ cfu of *B. thailandensis*, and evaluated a concentration range from 0--1000 µM. Growth was detected by absorbance at 600 nm using an Infinity M200 Pro plate reader (Tecan). Synergistic interactions of chloroxine and ceftazidime were tested by mixing equal volumes of media prepared using the micro dilution method, to test concentrations of each antibiotic from 0–32 µg/ml. Samples were then treated as above.

## $IC_{50}$ determination

90 µL of a culture prepared as above and resuspended in M9 media supplemented with 730 µM ceftazidime to an $OD_{600\ nm}$ of 0.8 was treated with two-fold dilutions of compounds in DMSO (in a final DMSO concentration of 0.016% (v/v)). Samples were incubated in a 96 well black walled plate covered with AeraSeal film (SigmaAldrich) at 28˚C for 24 hours before quantification of viable cells with PrestoBlue as above. $IC_{50}$ values were fitted to Eq (2) using Graphpad Prism version 6.0.1.

## Time dependent killing

A stationary phase culture of *B. thailandensis* was centrifuged and resuspended in 10 ml LB to an $OD_{600\ nm}$ of 0.4 (equivalent to $4 \times 10^8$ cfu/mL). Samples were treated with 730 µM ceftazidime hydrate, 30 µM chloroxine in DMSO, or both. Samples were incubated at 37˚C with shaking. 1 mL samples were taken at time intervals over a 48 hour period (0, 4, 8, 24, 48 hr), cells harvested and resuspended in LB before serial dilution and plating on agar. Colonies were counted following 24 hours incubation at 37˚C. All samples contained DMSO at 0.083% (v/v).

## Results

### Assay development

The aim of this study was to identify compounds that may have use as co-therapies for the treatment of infection with *B. pseudomallei*. We aimed to develop an assay that would identify compounds that reduced the proportion of *B. thailandensis* cells that remained viable when delivered in combination with ceftazidime (730 μM; 400 μg/ml; this is equivalent to the peak blood concentration given for melioidosis septicemia; c.f. MIC of 2–6 μg/ml for common *B. pseudomallei* isolates [38–40]). We evaluated the effectiveness of our assays using the Z' statistic [34], commonly used for high-throughput screening (HTS) [41]. The requirement of this initial assay was to distinguish a ceftazidime treated culture from heat killed cells (negative control) with a Z' score greater than 0.4. We investigated a range of absorbance, fluorescence, luminescence, and qPCR-based assays for correlates of cell viability (S1 File). A phenotypic cell viability assay using the resazurin-based reagent PrestoBlue met our criterion, showing a greater Z' score than the alternative assays (Fig 1). This cell viability assay offered high throughput screening that was convenient and affordable, with good discrimination in the reduction of surviving *B. thailandensis* cells. The assay conditions (incubation temperature, volume, and sealing) were then optimized for use in 384 well plates with automated dispensing of reagents. The final assay quality was determined, comparing *B. thailandensis* at the optimized cell density in M9 media supplemented with ceftazidime, to media without bacteria. The final assay resulted in a Z' score of 0.68 (S1 Fig), which is consistent with the HTS requirement for a score of 0.5–0.7. Assay quality was maintained throughout the HTS.

**High throughput screening.** Primary single point screening took place for the 61,250 compounds comprising the DDU's Diversity screening library [31]. As expected, the majority of the compounds were inactive (Fig 2), with some compounds showing compound effects on the assay (indicated by the tail of compounds showing >150% of the mean fluorescence). Using the median percentage effect plus three standard deviations, a cut-off of 34.3% inhibition was determined. This identified 2,127 unique compounds as 'hits', which exceeded the capability for downstream analysis. As a result, a pragmatic cut off of 45% inhibition was selected (Fig 2, red arrow), resulting in 345 unique compounds. Some of these were excluded due to known promiscuity issues. We selected 309 compounds for detailed screening: these included some near analogues to hits from within the DDU collection that showed good activity and physiochemical properties. For these 309 compounds, a ten point, 2-fold, concentration response assay was performed in duplicate (S2 Fig) with criteria for a positive hit set at greater than 50% inhibition at the highest concentration tested (100 μM). Acceptable concentration response relationships were returned for 58 compounds, of which 29 showed a $p$IC$_{50}$ (-log$_{10}$(IC$_{50}$)) values > 5, indicating 50% activity at 10 μM and a potential "hit".

*Down selecting compounds.* Concentration dependent killing assays were repeated for the 29 compounds selected using newly sourced stocks of the same compounds and performed in triplicate over a larger range of concentrations. $p$IC$_{50}$, hillslope and maximal effect were used to further down select to six compounds (**A-F**), all of which displayed a $p$IC$_{50}$ > 5 in either the primary assay (Fig 3A, Table 1), the secondary assay (S3 Fig, S1 Table) or both. One of these compounds (compound **A**, 5,7-dichloroquinolin-8-ol, also known as chloroxine; Fig 3B), is a currently licensed antimicrobial. Our further investigations focused on chloroxine.

*Minimum inhibitory concentration (MIC).* The experiments described above highlighted that chloroxine had activity on cells that had survived following ceftazidime treatment. Our hypothesis was that chloroxine either potentiated the effects of ceftazidime or was toxic to cells in a metabolic state that rendered them insensitive to ceftazidime. We reasoned chloroxine might be acting as an antibiotic in its own right, as so determined its MIC. Chloroxine

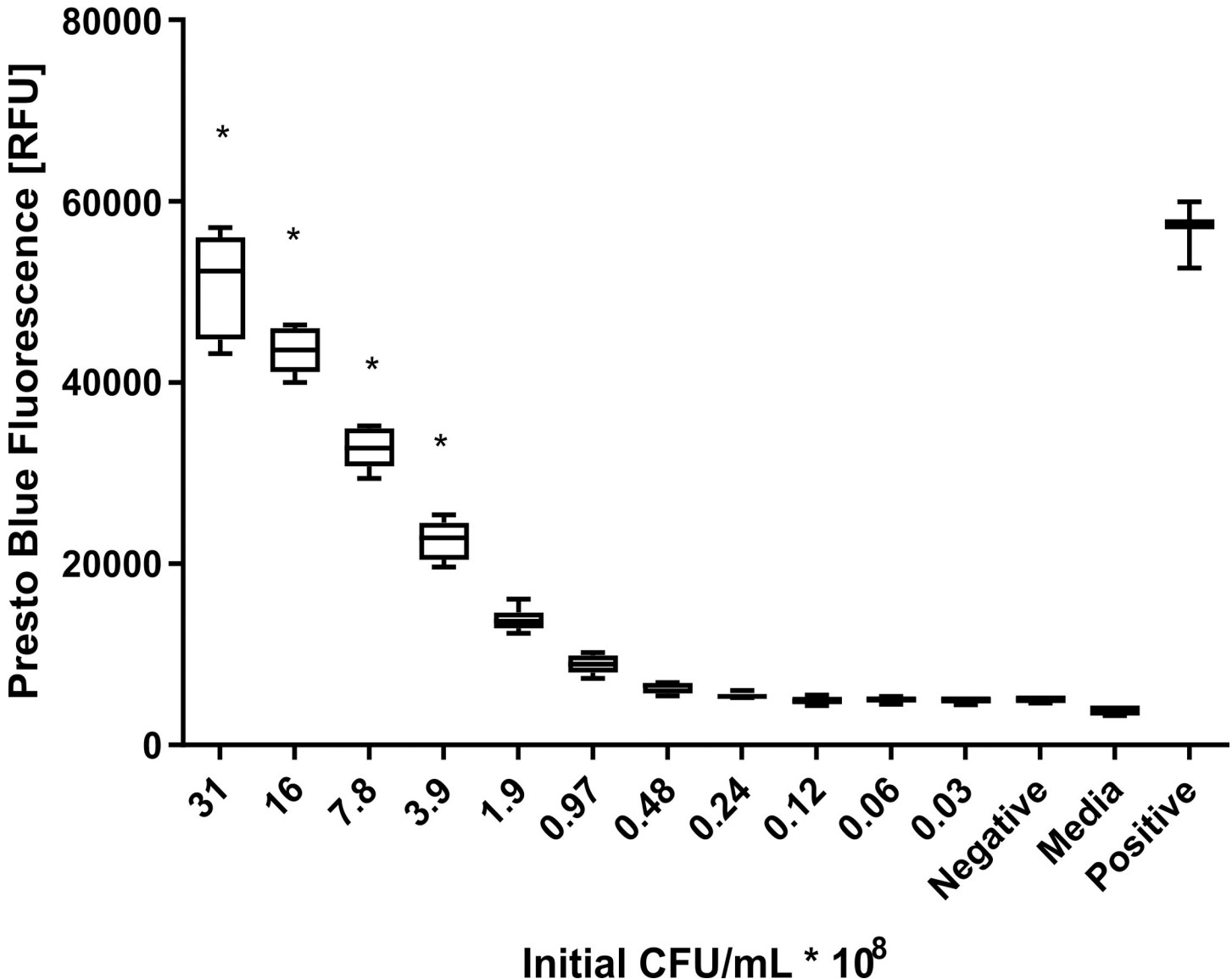

**Fig 1. The PrestoBlue assay shows discrimination between the numbers of surviving cells.** A *B. thailandensis* culture was harvested, resuspended, and diluted in M9 media supplemented with 730 μM ceftazidime, to provide a series of cell densities at two-fold intervals. Samples were incubated statically at 28°C in 96 well plates. PrestoBlue was added following 20 hours of incubation and the fluorescence read gain optimized for the highest bacterial concentration. The results show reliable discrimination of two-fold differences in cell numbers when compared to a heat killed cell negative control. * All show Z' > 0.5 when compared to the controls. Positive control: Cells resuspended in M9 media without ceftazidime. Data shows biological triplicates, whiskers indicate the minimum and maximum results, the box the 25th to 75th percentiles and the central line indicates the median.

demonstrated antimicrobial activity, with an MIC of 4 μg/mL (compared with 4–8 μg/mL for ceftazidime). This was not unexpected, as chloroxine is known to be an effective antimicrobial with activity described against a range of Gram-positive bacteria and fungi. Chloroxine and ceftazidime showed no evidence of synergistic effects on MIC (S4 Fig), suggesting that the effects observed reflected the potentiation of the ceftazidime effect on tolerant cells.

*Time dependent killing*. A time dependent killing assay was performed to demonstrate that the effect of chloroxine was complementary to ceftazidime. Stationary phase cells were resuspended in fresh media supplemented with ceftazidime (100X MIC), with or without 30 μM of chloroxine. Bacterial counts were determined over 48 hours of incubation. Chloroxine

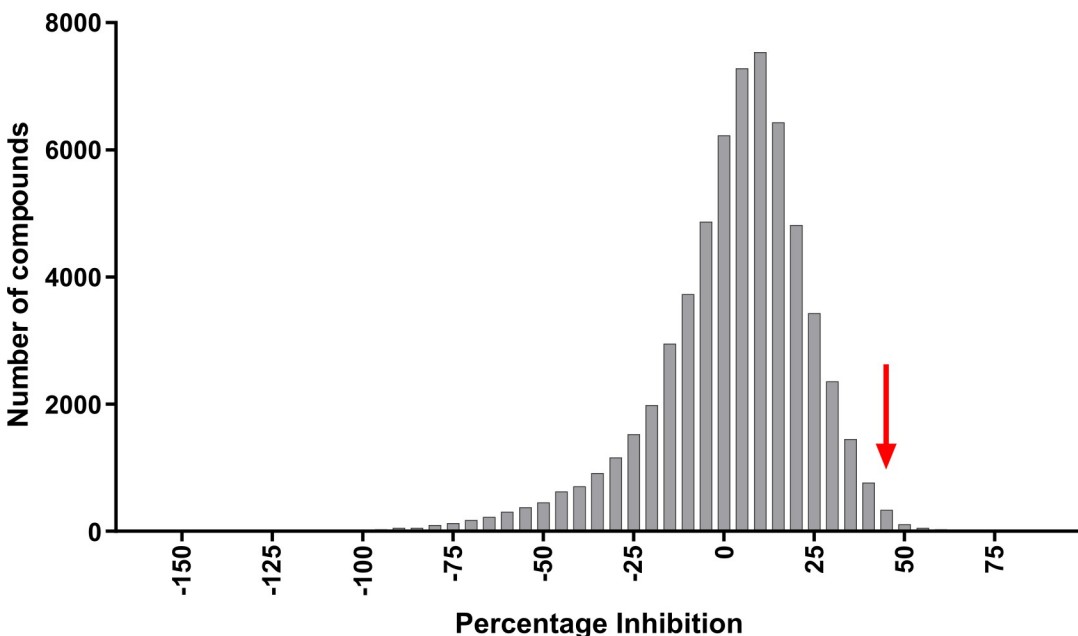

**Fig 2. Inhibitory activity of test compounds screened with a phenotypic assay.** A *B. thailandensis* culture was harvested and resuspended in M9 media supplemented with 730 µM ceftazidime and 30 µM of each of the compounds. Cells were grown at 28˚C for 20 h, and PrestoBlue added. Inhibition was calculated by comparisons to the controls. The distribution shows the percentage inhibition grouped in 5% windows for the HTS of 61,250 compounds. The median activity is 5.3%. The standard deviation of the positive tail is 9.65%, giving a statistical cut-off for activity of 34.3%. The red arrow indicates the selected pragmatic threshold at 45%. 345 compounds were identified as 'hits' according to this criterion.

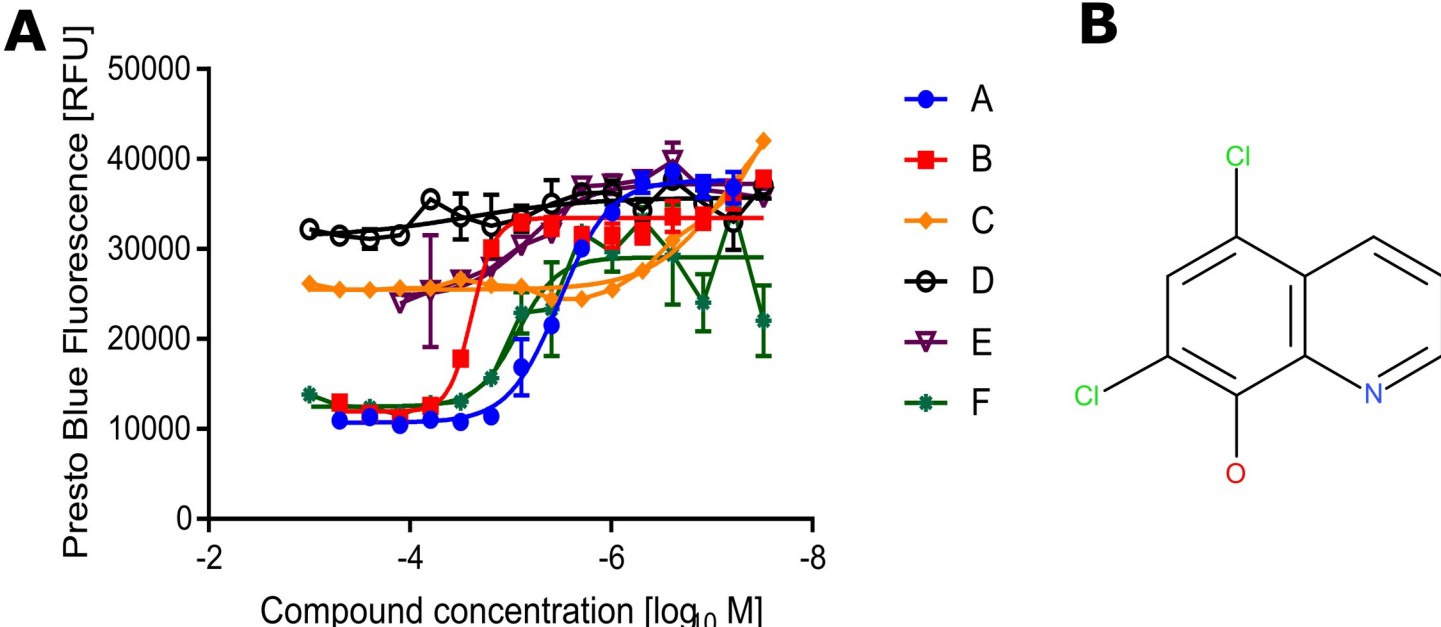

**Fig 3. $p$IC$_{50}$ determination of six candidate compounds using the PrestoBlue cell viability assay. A**: A *B. thailandensis* culture was harvested and resuspended to a concentration of $8 \times 10^8$ CFU/mL in M9 media supplemented with 730 µM ceftazidime. This was added to a 96 well plate containing two-fold dilutions of compounds in DMSO from 500 µM. Plates were incubated for 24 hours at 37˚C before the addition of the PrestoBlue cell viability reagent and the fluorescence read. Results show three biological replicates with error bars indicating standard error. The derived IC$_{50}$ values are detailed in Table 1. **B**: Structure of chloroxine.

**Table 1. Analysis of the concentration dependent killing data shown in Fig 3.**

| Compound | Top/RFU | Bottom/RFU | Hill Slope | $p$IC$_{50}$ (IC$_{50}$ in M) | R Square | Number of Points Analysed |
|:---:|:---:|:---:|:---:|:---:|:---:|:---:|
| A | 37600 | 10700 | 1.7 | 5.5 | 0.98 | 42 |
| B | 33400 | 11900 | 3.8 | 4.6 | 0.95 | 45 |
| C | 48000 | 25500 | 1.1 | 7.2 | 0.96 | 48 |
| D | 32000 | 36000 | 0.7 | 4.6 | 0.22 | 48 |
| E | 25000 | 37200 | 1.5 | 5.2 | 0.75 | 48 |
| F | 12000 | 29000 | 2.2 | 5.1 | 0.72 | 48 |

significantly reduced the number of viable cells following incubation for 24 hours when compared to treatment with ceftazidime or chloroxine ($p < 0.05$, and $p < 0.005$ respectively), with a reduction in cell number by nearly two orders of magnitude at 48 hours (Fig 4). This validated the antimicrobial activity of chloroxine. We also performed a cytotoxicity assay that demonstrated that chloroxine was not toxic to mammalian cells (S5 Fig).

*Hit expansion.* One possibility was that chloroxine was acting non-specifically as an oxidizing agent. Hit expansion using similar commercially available compounds was performed to gain insight into the structure-activity relationship. This would also assist in the future development of this compound from hit to lead.

Chloroxine (Fig 3B) is a small synthetic compound with limited scope for improvement. The $p$IC$_{50}$ was determined as 5.5 using the PrestoBlue assay (Fig 3A). A total of eleven similar compounds were commercially available and were used for this screen. None of these demonstrated increased potency in the assay (Fig 5). However, the pattern of loss of potency provides clear insights into how chloroxine could be further modified. It was clear that the identity of the substituent at the 7-position was important. Replacement of this with an amino group significantly reduced activity ($p$IC$_{50}$ reducing to 3.4; Fig 5A). Similarly, addition of a methyl group at the 2-position was poorly tolerated, leading to a loss of detectable activity at the concentrations tested (Fig 5B). The halogens in the compound could also be altered to some extent. Replacement of the chlorine atom with bromine at the 7-position was tolerated, but only if the chlorine in the 5-position was also removed (reduction in $p$IC$_{50}$ from 5.5 to 5.1, Fig 5C). Two alternative structures with bromine were not active (S2 Table). Iodide ions were also tolerated in place of the chlorines, again with a small loss of activity (S2 Table). More extensive alterations to the structure of chloroxine resulted in the loss of at least one order of magnitude of activity (S2 Table).

Finally, to validate the use of *B. thailandensis* as a proxy for *B. pseudomallei*, we repeated the original PrestoBlue assay with ceftazidime and chloroxine against *B. pseudomallei*. The fluorescent signal seen for *B. pseudomallei* was approximately double the *B. thailandensis* signal; this is unlikely to be significant as the level of fluorescence is known to vary between species with this reagent [42]. Chloroxine demonstrated a similar level of activity against *B. pseudomallei* to that seen against *B. thailandensis* (Fig 6; IC$_{50}$ for *B. thailandensis* 2.0 μM (95% CIs 1.7–2.2 μM); IC$_{50}$ for *B. pseudomallei* 9.2 μM (95% CIs 7.2 to 12 μM). This result suggests that our assay could be used with *B. pseudomallei*.

## Discussion

This study aimed to identify compounds that were effective in reducing the proportion of *Burkholderia* cells that survive following treatment with ceftazidime at concentrations much higher than the MIC. Ceftazidime is a front-line therapy for the acute phase of the disease melioidosis [1]. Ceftazidime specifically targets penicillin-binding protein 3 in *B. pseudomallei*

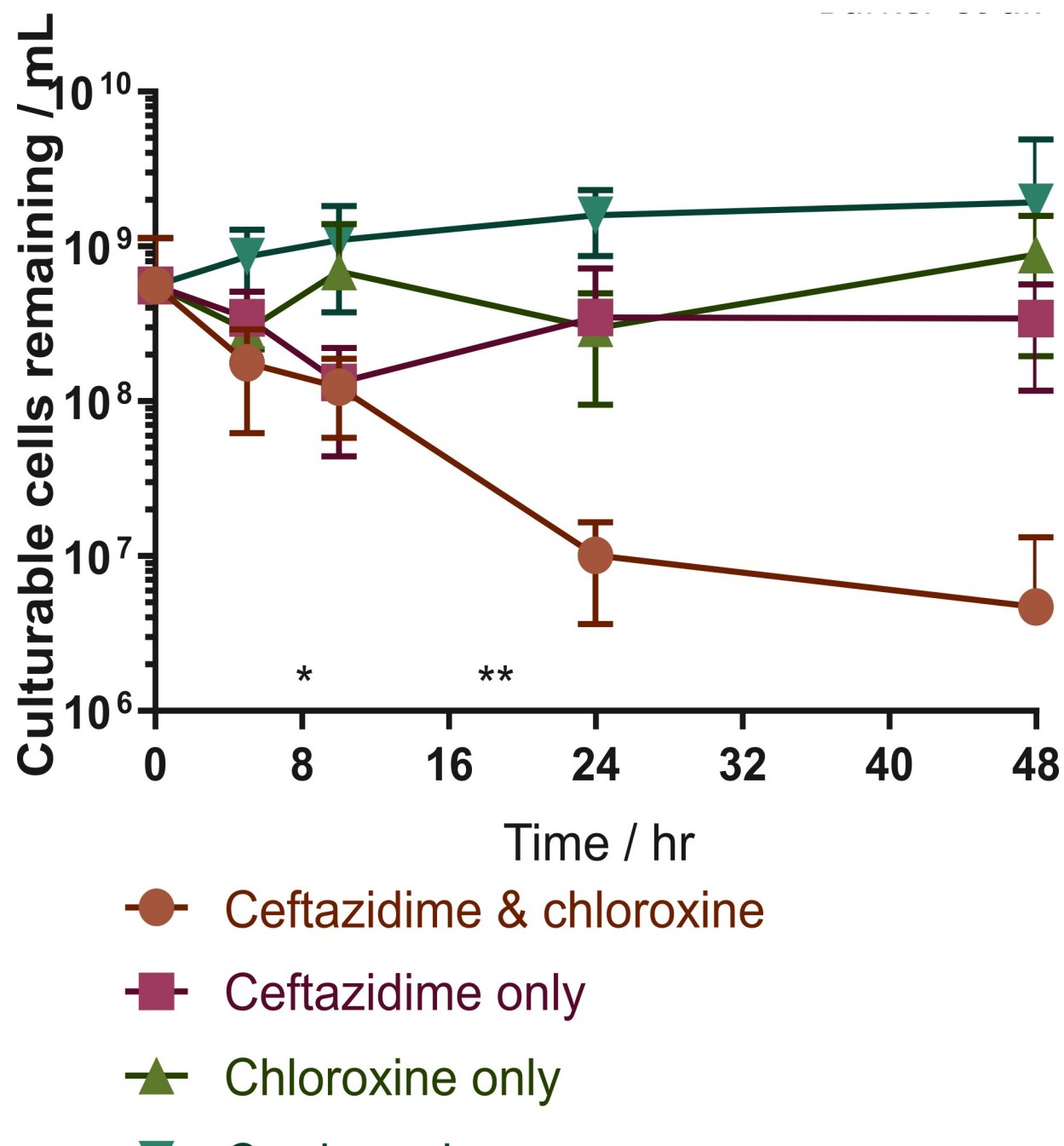

**Fig 4. A secondary assay evaluating the number of culturable cells remaining following treatment with ceftazidime and chloroxine.** A culture of *B. thailandensis* was treated with 730 μM ceftazidime hydrate, 30 μM **chloroxine**, both, or neither. Samples were incubated at 37°C with shaking. Samples were taken at time intervals, cells harvested and resuspended in LB broth before serial dilution and enumerating on agar. Error indicates standard error of serial dilution and CFU count. $n = 6$. Differences between the ceftazidime alone, **chloroxine** alone, and ceftazidime and **chloroxine** samples were analyzed using a Kruskal-Wallis test with Dunn post-hoc comparison using Graphpad v. 7.03. *— $p < 0.01$ between **chloroxine** alone, and ceftazidime with **chloroxine**. **— $p < 0.05$ between both **chloroxine** alone and ceftazidime alone, and both compounds.

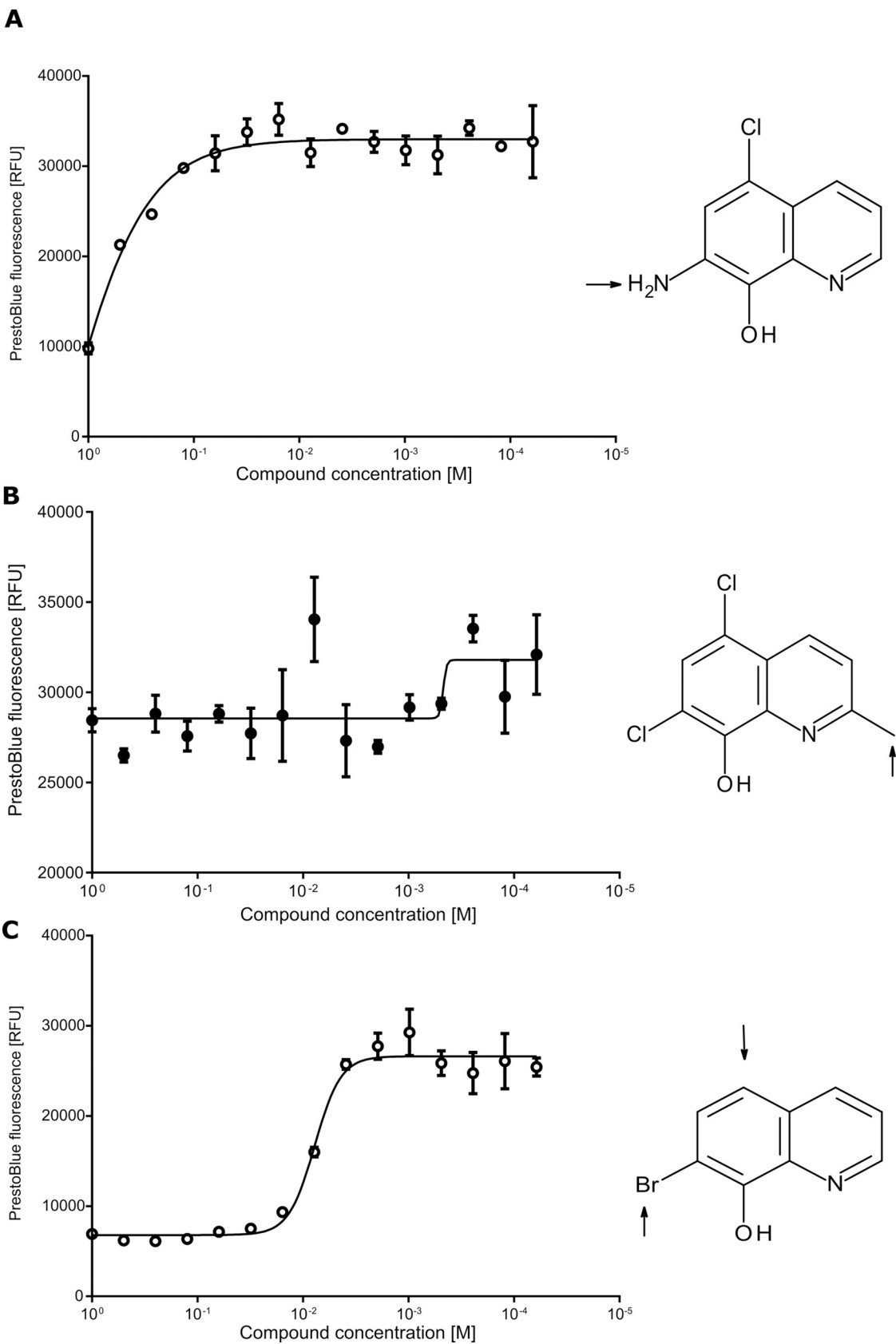

**Fig 5. Hit expansion around chloroxine.** A *B. thailandensis* culture was harvested and resuspended to a concentration of $8x10^8$ CFU/mL in M9 media supplemented with 730 μM ceftazidime. This was added to a 96 well plate containing two-fold dilutions of compounds in DMSO from a starting concentration of 1 mM. Plates were incubated for 24 hours at 37°C before addition of the PrestoBlue cell viability reagent and the fluorescence read. Results show three biological replicates with error bars indicating standard deviation. These experiments are equivalent to those in Fig 3 and can be compared to chloroxine in Fig 3. 7-amino-5-chloro-8-quinolinol differs to chloroxine through substitution of an amino group for a chlorine at position 7 (A). This modification causes a significant decrease in this compound's activity as a co-treatment with ceftazidime, with a $p$IC50 ≈ 3.4. 5,7-dichloro-2-methyl-8-quinolinol differs from chloroxine by addition of a methyl group in the 2-position (B). This addition abolishes this compound's activity as a co-treatment with ceftazidime at the concentrations tested. 7-Bromo-8-quinolinol differs from chloroxine by the removal of chlorine at the 5-position, and replacement of chlorine by bromine at the 7-position (C). This compound retains activity as a co-treatment with ceftazidime that is comparable with the parent compound ($p$IC50 = 5.1, 5.5 for chloroxine).

[23,43]; the *B. thailandensis* orthologue shows 97% identity at the amino acid level. Ceftazidime treatment at sub-MIC concentrations induces filamentation of *B. pseudomallei*, whilst ceftazidime is lytic at higher concentrations [44]. Our study aimed to identify compounds that could be developed to be administered alongside front-line treatments, with the aim of reducing the rates of recurrent infection. This may then allow the duration and cost of the treatment to be reduced.

We decided that the use of a whole cell, phenotypic assay was advantageous for this application. Cells in the low metabolic state that provide resistance to antibiotics such as ceftazidime are heterogeneous [45], and cover a range of phenotypes. As such, a phenotypic assay focusing

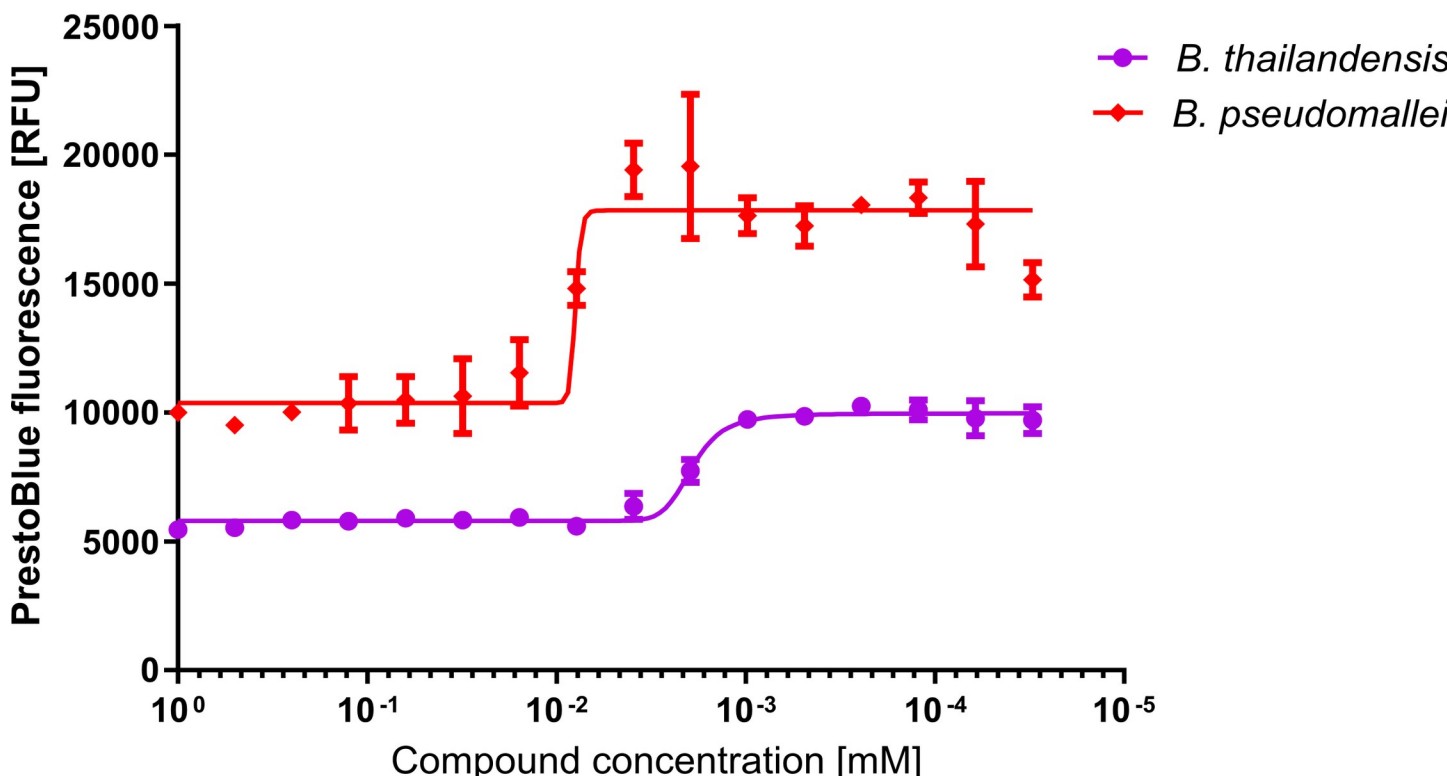

**Fig 6. $p$IC$_{50}$ determination using the PrestoBlue cell viability assay to compare the concentration dependent killing for ceftazidime used in combination with chloroxine to treat *B. thailandensis* and *B. pseudomallei*.** *B. thailandensis* and *B. pseudomallei* cultures were harvested and resuspended to a concentration of $8x10^8$ CFU/mL in M9 media supplemented with 730 μM ceftazidime. These were added to a 96 well plate containing two-fold dilutions of chloroxine in DMSO. Plates were incubated for 24 hours at 37°C before the addition of the PrestoBlue cell viability reagent and determination of fluorescence. Results show three biological replicates with error bars indicating standard deviation. $p$IC$_{50}$ for *B. thailandensis* = 5.7; $p$IC$_{50}$ for *B. pseudomallei* = 5.0.

on the reductive state of the cell was preferred over a target-based assay for identifying tractable hit compounds. In addition, phenotypic screening is regaining popularity over target-based screening. The principal reasons for this are that compounds with the physiological ability to penetrate Gram-negative cells, function *in vivo* and avoid efflux pumps are identified [46,47]. Consequently, the compound series obtained are likely to have significant advantages for downstream optimization and development. The use of cell viability reagents, allowing the assessment of cells at a population or individual level, offered the opportunity to identify viable cells in a variety of states, and was considered more relevant for this work. Resazurin based assays have previously been shown to identify all viable cells, and not just the less abundant "persister" cells [48]. Other phenotypic assays that have identified co-therapies against other organisms have exploited colony counting [49,50], DNA binding dyes [51] and Live/Dead reagents [52]. The PrestoBlue resazurin-based assay proved effective, in *Burkholderia*, at identifying compounds that were active at concentrations below 10 μM, validating the approach. Primary screening with the DDU's diversity library identified 2,127 compounds that showed a significant effect as a co-therapy with ceftazidime, based on an activity threshold of 34.3% inhibition. The preliminary hit rate was 3.5%, which is in the expected range for an effective assay.

Chloroxine was identified as a potential co-therapy to treat infection with *B. thailandensis*. This compound demonstrated strong activity in the primary assay ($IC_{50}$ = 2 μM) and resulted in a significant reduction in the number of culturable bacterial cells following 24–48 hours treatment, in combination with ceftazidime (>100-fold reduction). This is similar to the level of efficacy that has been previously observed with compounds targeting *E. coli* [51]. It is hypothesized that chloroxine would reduce the proportion of cells surviving ceftazidime treatment, and so reduce the intensive treatment phase in patients with melioidosis. Chloroxine has known bacteriostatic, fungistatic and antiprotozoal properties [53] and has previously been shown to have synergistic effects with minocycline against *Pseudomonas aeruginosa* [54]. Consistent with this, administration of chloroxine alongside the frontline treatment for melioidosis, ceftazidime, demonstrated improved activity than the compounds evaluated as sole therapies. This suggests that the compounds have complementary effects when treating *B. thailandensis*. Bactericidal effects were observed at concentrations below the chloroxine MIC (Figs 3A and S4). This study demonstrates evidence for the concept of use of chloroxine as a complementary agent to ceftazidime against *B. thailandensis*.

Hit expansion was carried out for chloroxine. Only a limited range of compounds around the chloroxine structure were available. None of the compounds evaluated demonstrated improved activity compared to the parent compound (S2 Table). However, it became evident that only limited substitutions at the chlorine positions were tolerated, only compounds with other halides in these positions showed comparable activity to chloroxine. Furthermore, addition of a methyl group in the 2-position was sufficient to abolish activity at the concentrations evaluated (Fig 5). These data strongly suggest that chloroxine has some specificity, and it is not a consequence of its suggested oxidative activity. The addition of a methyl group would not be expected to reduce the oxidative capability of chloroxine, yet this abolishes activity. Furthermore, the iodo-equivalent of chloroxine retains similar activity to chloroxine and is considerably less oxidizing. This hit expansion validates the hypothesis that chloroxine acts specifically to potentiate the effect of ceftazidime.

*B. thailandensis* was used in the preliminary experiments as a surrogate for *B. pseudomallei*. Evaluation of chloroxine against *B. pseudomallei* showed that chloroxine is effective as a co-therapy for ceftazidime at druggable concentrations (Fig 6). Although activity is reduced compared to that observed against *B. thailandensis*, these results validate the use of *B. thailandensis* as a surrogate in this study. In the context of ongoing treatment for cutaneous melioidosis, chloroxine is currently licensed for topical treatment of skin infections. It may become a useful

addition to the existing portfolio of treatments for cutaneous melioidosis due to its low cost and activity against *B. pseudomallei*. Determining the efficacy against a wider range of strains of *B. pseudomallei* would provide further confidence to this proposed use. Development of co-therapies suitable for systemic treatment would require significant chemical modification to optimize activity and bioavailability. Chloroxine is soluble to 644 µM in water, and the peak ceftazidime concentration in serum is 130 µM [55]. Furthermore, *B. pseudomallei* invades and multiplies in phagocytic cells [56], so modification for penetration of these cells would be necessary. A wider range of starting lead scaffolds would likely be necessary for such optimization.

## Conclusions

Our study has demonstrated that a phenotypic assay can identify compounds that act as cotherapies for frontline antibiotics in *Burkholderia*. A high throughput screen of 61,250 compounds identified six compounds that demonstrated activity at concentrations of less than 10 µM. One of these compounds, 5,7-dichloro-8-quinolinol (chloroxine), is currently licensed for other indications. Although hit expansion with commercially available compounds did not identify any neighbors with improved activity, chloroxine significantly reduced the number of surviving cells over 48 hours. Our data suggest that similar approaches could be highly efficacious in identifying useful compounds for use with other bacteria with similar clinical challenges.

## Supporting information

**S1 Fig. A checkboard of microbial culture to show positional plate effects.** A *B. thailandensis* culture was harvested and resuspended to a concentration of $8\times10^8$ CFU/mL in M9 media supplemented with 730 µM ceftazidime. 45 µl of this suspension (green) and a heat killed control (red) were added to each well in quarters of a 384 well plate. Samples were incubated statically at 28˚C. After 20 hours, PrestoBlue was added and the fluorescence read. Intensity of colour indicates the signal strength. Maximum signal variance was 11.2%CV, with Z' = 0.68 (Calculation: Mean of positive wells = 204,371, SD = 15,179; mean of negative wells = 37,339, SD = 2,532). Relative fluorescence units (RFU) are given for all wells showing significantly decreased fluorescence in edge and corner wells compared to central wells ($p = 0.011$). Calculation of derived Z (as a worked example of all such calculations in the manuscript):
Positive wells: Mean ($\mu_p$) = 204371, standard deviation ($\sigma_p$) = 15179.
Negative wells: Mean ($\mu_n$) = 37339, standard deviation ($\sigma_n$) = 2532.
Difference of means: $\mu_p$—$\mu_n$ = 204371–37339 = 167032.
Sum of standard deviations: $\sigma_p + \sigma_n$ = 15179 + 2532 = 17711.

$$Z\ Factor = 1 - \frac{3(\sigma p + \sigma n)}{|\mu p - \mu n|}$$

Z = 1 –(3 * 17711/167032)
Z = 1–0.318
Z = 0.68.
(TIF)

**S2 Fig. A *B. thailandensis* culture was harvested and resuspended to a concentration of $8\times10^8$ CFU/mL in M9 media supplemented with 730 µM ceftazidime.** This was added to a 96 well plate containing a concentration response assay performed in duplicate two-fold dilutions of compounds in DMSO. Plates were incubated for 24 hours at 37˚C before addition of

PrestoBlue and the fluorescence read. The criterion for a positive hit was set as greater than 50% inhibition at the highest concentration tested (100 μM).
(TIFF)

**S3 Fig. $pIC_{50}$ determination from compounds B-F using SYTO9.** The Live/Dead reagent SYTO9 was used to quantify viability as a function of the membrane integrity of the cell. A *B. thailandensis* culture was harvested and resuspended to a concentration of $8x10^8$ CFU/mL in M9 media supplemented with 730 μM ceftazidime. This was added to a 96 well plate containing two-fold dilutions of compounds in DMSO. Plates were incubated for 24 hours at 37°C before addition of the Live/Dead cell viability reagents and the fluorescence read. Results show three biological replicates with error bars indicating standard error. The derived $IC_{50}$ values are shown in S1 Table.
(TIFF)

**S4 Fig. Synergistic effect study.** A *B. thailandensis* culture was diluted to an $OD_{600}$ of 0.004 in Muller-Hinton broth (MHB; Sigma). Solutions of ceftazidime and chloroxine at 4X final concentration in MHB were prepared by serial dilution from a master stock. Stocks were mixed one part chloroxine stock, one part ceftazidime stock, and two parts *B. thailandensis* culture (giving an inoculum of ~5 x $10^5$ cfu) in a 96 well plate. Samples were sealed and grown at 37°C statically for 20 hr, following which absorbance at 600 nm was read using a plate reader. Values were corrected for non-inoculated controls. Wells that showed growth ($OD_{600} > 0.1$, corresponding with the results of visual inspection; no antibiotic controls showed an $OD_{600}$ of $0.88 \pm 0.1$, $n = 8$) are highlighted in red. The plate reader results were in correspondence with visual inspection.
(TIFF)

**S5 Fig. Chloroxine does not show cytotoxic effects.** Chloroxine was tested to determine whether it had any cytotoxicity against mammalian cells. Neuroblastoma cells were selected as a representative mammalian cell line that is robust and unaffected by DMSO at concentrations up to 1% (v/v). Cells were plated at 20,000 cells/well in 100 μl Dulbecco's media. 300 μM chloroxine in 0.5% (v/v) DMSO, or 0.5% (v/v) DMSO (carrier) was added, and the plate incubated for 4 or 24 hours. Cytotoxicity was determined using an LDH cytotoxicity assay kit (Thermo Scientific #88953). Briefly, 10 μl of lysis solution (to indicate 100% lysis) or water (control) was added to untreated wells, and these incubated at 37°C for 45 min. 50 μl of supernatant from each well was added to 50 μl of room temperature assay solution in a 96 well plate (Greiner Bio-One #655201). Samples were incubated at room temperature in the dark for 30 min, and 50 μl of assay stop solution added. Absorbance at 490 nm and 680 nm was read in a M200 Pro plate reader (Tecan), with the difference between these representing LDH activity. % cytotoxicity was determined on a linear scale between the measurements for 100% lysis and water only control. No significant difference was observed between treated and control cells (two-way ANOVA testing for effect of compound or time gives $p > 0.5$ for each effect). $n = 6$; image shows means with error bars showing SEM.
(TIFF)

**S1 Table. Activities of the six most promising compounds.** The structures, derived $IC_{50}$ values from the resazurin and SYTO9 based assays and the MIC values are provided for each compound. 95% confidence intervals are shown in parenthesis. $IC_{50}$ values and confidence intervals were calculated using Graphpad v. 8.3. MIC values were determined as the lowest concentration not showing significant growth.
(DOCX)

**S2 Table. Hit expansion structures and activity for chloroxine.** Each compound was tested using the PrestoBlue assay. A *B. thailandensis* culture was harvested, and resuspended to a concentration of $8x10^8$ CFU/mL in M9 media supplemented with 730 μM ceftazidime. This was added to a 96 well plate containing two-fold dilutions of compounds in DMSO. Plates were incubated for 24 hours at 37˚C before the addition of PrestoBlue and the fluorescence read. Results show three biological replicates with error bars indicating standard deviation. All modifications resulted in reduced activity when compared to chloroxine. In cases where the data did not fit to the model used (where no activity is demonstrated at the concentrations used), $p$IC$_{50}$ is recorded as N/A.
(DOCX)

**S1 File. Supplementary results.** A series of approaches were trialed to identify an effective assay for determining the level of *B. thailandensis* cells surviving following 24 hours of exposure to 730 μM ceftazidime. The PrestoBlue approach that was eventually selected is described in detail in the main paper. The criteria used for selection was the ability to identify a four-fold difference in initial cell numbers with clear statistical significance; affordability of reagents for over 60,000 test samples; and ease of use in a high throughput setting.
(PDF)

# Acknowledgments

The authors thank Dr. Claudia Hemsley and Professor Rick Titball (University of Exeter, UK) for advice and assistance during the development of these assays, and Dr. Akshay Bhinge (University of Exeter, UK), for assistance with the cytotoxicity assay.

# Author Contributions

**Conceptualization:** Sarah V. Harding, David Gray, Helen S. Atkins, Nicholas J. Harmer.

**Formal analysis:** Sam Barker, Nicholas J. Harmer.

**Investigation:** Sam Barker, Mark I. Richards, Nicholas J. Harmer.

**Methodology:** Nicholas J. Harmer.

**Supervision:** Sarah V. Harding, David Gray, Helen S. Atkins, Nicholas J. Harmer.

**Writing – original draft:** Sam Barker, Sarah V. Harding, David Gray, Helen S. Atkins, Nicholas J. Harmer.

**Writing – review & editing:** Sarah V. Harding, Nicholas J. Harmer.

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
