## [Decision Letter · Decision Letter 0]

15 Oct 2020

PONE-D-20-28523

Drug screening to identify compounds to act as co-therapies for the treatment of Burkholderia species

PLOS ONE

Dear Dr. Harmer,

Thank you for submitting your manuscript to PLOS ONE. After careful consideration, we feel that it has merit but does not fully meet PLOS ONE’s publication criteria as it currently stands. Therefore, we invite you to submit a revised version of the manuscript that addresses the points raised during the review process. 

Both reviewers agreed that this version was greatly improved from the initial version.  However, there are just a few questions that still need to be addressed, which have been listed clearly from Reviewer 1.

We look forward to receiving your revised manuscript.

Kind regards,

R. Mark Wooten, Ph.D.

Academic Editor

PLOS ONE

Journal Requirements:

We note that one or more of the authors are employed by a commercial company: Phoremost Ltd.

2.1. Please provide an amended Funding Statement declaring this commercial affiliation, as well as a statement regarding the Role of Funders in your study. If the funding organization did not play a role in the study design, data collection and analysis, decision to publish, or preparation of the manuscript and only provided financial support in the form of authors' salaries and/or research materials, please review your statements relating to the author contributions, and ensure you have specifically and accurately indicated the role(s) that these authors had in your study. You can update author roles in the Author Contributions section of the online submission form.

2.2. Please also provide an updated Competing Interests Statement declaring this commercial affiliation along with any other relevant declarations relating to employment, consultancy, patents, products in development, or marketed products, etc.  

Reviewers' comments:

Reviewer's Responses to Questions

**Comments to the Author**

1. Is the manuscript technically sound, and do the data support the conclusions?

Reviewer #1: Yes

Reviewer #2: Yes

2. Has the statistical analysis been performed appropriately and rigorously? 

Reviewer #1: I Don't Know

Reviewer #2: Yes

3. Have the authors made all data underlying the findings in their manuscript fully available?

Reviewer #1: Yes

Reviewer #2: Yes

4. Is the manuscript presented in an intelligible fashion and written in standard English?

Reviewer #1: Yes

Reviewer #2: Yes

5. Review Comments to the Author

Reviewer #1: I am Reviewer #2 from the prior submission. I have read and appreciate the responses made by the authors to my comments and concerns. I have also read the insightful comments and concerns of Reviewer #1 from the prior submission and I am generally satisfied that the authors have addressed those points. The thoughtful discussion contributions made by both parties are very much appreciated.

It is my view that the text of the paper alone reads well at present. It is quite straightforward and I believe that the current version is better than the last.

However, I am frustrated about the way the Figures and Tables (including those that are Supplementary) are integrated with one another and the text. In order to follow along, one must open the Results section of the text and open the corresponding (in-text) Figures/Tables and also have the Supplementary Figures/Tables open as well as the Experimental Procedures section of the text to gather all of the information needed to understand a given result. This frustration has been mentioned by other, prior Reviewers (see discussions pertinent to readability and sequence of data presentation). This frustration seems to extend beyond the routine and expected awkwardness of navigating a manuscript before it has been accepted and transformed into the final publication format. It makes me wonder if a full re-working of the manuscript is necessary rather than merely updating it piecemeal to satisfy the comments garnered in each additional round of review. No doubt the document has been reviewed and updated many times.

A key concern I have is that the authors seem to refer to elements of figures (particularly Supplementary Figures), which I cannot find. This makes me wonder if I am looking at the same version of the Supplementary Figures as the authors are looking at. Please see below for line-by-line examples.

I thus conclude that from a scientific standpoint, I am satisfied that my concerns have been addressed. However, I’d encourage the authors to sit with the paper and take a very careful look at all of the references to in-text / Supplementary Figures and Tables to make sure that everything is coherent.

Line-by-line Comments

Lines 95-99: “Previous studies have shown that approximately 0.1% of B. thailandensis cells survive for 24 hours following treatment with 100X MIC.” Likewise, can you quote a survival percentage for B.P. from the literature – if available?

Line 166-169: “pIC50, hillslope and maximal effect were used to further down select to six compounds (A-F), all of which displayed a pIC50 > 5 (Figure 3A, Table 1, Supplementary Figure 8, Supplemental Table 1)”. Firstly, it is tremendously confusing to make a statement and then provide substantiation of the statement by referring to an in-text figure, an in-text table, a supplementary figure and a supplementary table. It is almost enough to cause the reader to give up on trying to independently review points of evidence for the claim. There must be a better way to convey which parts of the claim are substantiated by evidence in which in-text figures and tables and in which Supplementary figures and tables. Secondly, I don’t see the structures for Compounds A-F in Supplementary Table 1. Are they supposed to be here or elsewhere – for example in one of the other tables or figures? Moreover, which letter (A-F) is chloroxine in each figure/table?

Line 179-180: The text suggests that there is a depiction of chloroxine’s antimicrobial activity with an MIC of 4 ug/mL in Supplementary Table 2. I do not see any information regarding this in Supplementary Table 2. Yet again, there is discussion of Reviewer #1’s comments included in the information for the authors from the 2nd submission of this manuscript to PLOS ONE. Here, it sounds like the authors are directing Reviewer #1 to see Supplemental Table 1. I don’t see this information there either. Please address.

Figure 1: Please state what the positive control is in the figure legend. Otherwise one must dig and look in the Experimental Procedures section to find it.

Response to Reviewer#2 Comment from Prior Submission: Previously, I requested an example of how the ‘Z’ factor is calculated. In your response, you indicated that you have included one and I was eager to take a look at that. However, I cannot find an example in the legend of Supplementary Figure 1, which is where I was directed to look. This makes me question if you have perhaps submitted the wrong version of the Supplementary Figures and that I am consequently now looking at the wrong version (?). Perhaps I am not looking at the final draft of your updates to the Supplementary Figures (?). This could account for my ongoing confusion. For context, this is what I see:

Supplementary Figure 1: Assay development using Bactiter Glo to measure ATP levels.

ATP levels in an untreated culture of B. thailandensis was quantified with Bactiter Glo reagent, in a series of two-fold dilutions with media. Signal from the Bactiter Glo was converted to an ATP concentration using an ATP standard curve in the same media. Results are the mean of three replicates. Error indicates 95% confidence intervals. Z’ for 0.8 to 0.4 = 0.61.

Thank you for your careful consideration to these matters.

Reviewer #2: This reviewer acknowledges that the author and colleagues have addressed all the comments and concerns posed in the review, included pertinent data, added missing information and corrected typographical errors.

6. PLOS authors have the option to publish the peer review history of their article (what does this mean?). If published, this will include your full peer review and any attached files.

Reviewer #1: No

Reviewer #2: No

---

## [Author Response · Author response to Decision Letter 0]

17 Dec 2020

Editor comments:

1. The manuscript has been amended to meet PLOS ONE's style requirements.

2.1 An amended funding statement has been provided below. This was also supplied in the cover letter.

“The authors have declared that no competing interests exist. Following the conclusion of his experimental and writing work on this manuscript, author SB was employed by Phoremost Ltd (2015-2020) and Midatech Pharma Plc (2020). The funder provided support in the form of salaries for author SB but did not have any additional role in the study design, data collection and analysis, decision to publish, or preparation of the manuscript. The specific roles of these authors are articulated in the ‘author contributions’ section."

2.2 We provide an amended Competing Interests Statement, which was also provided in the cover letter.

“The authors note that author SB was employed by Phoremost Ltd and Midatech Plc after completing his work on this manuscript, but before the final manuscript was completed. These companies provided his salary but did not have any influence on the text or data of the manuscript, or the decision to publish. This does not alter our adherence to PLOS ONE policies on sharing data and materials. The authors declare no other competing interests.”

3. The data have already been submitted to our institutional repository and been checked and approved. Once the manuscript is accepted (i.e. once it is clear that no further changes are required), a doi and URL will be generated which I will communicate so that the data availability statement can be updated.

4. I have provided captions for the supporting information and figures as separate files in accordance with journal policy.

Response to reviewers (also available as a marked up document):

We thank the reviewers for their careful reading of the manuscript, and for the additional suggestions for further improvement. We have revised the manuscript to meet all of these comments.

We have also revised the manuscript as requested by PLoS One to meet the PLoS style.

Reviewer #1: I am Reviewer #2 from the prior submission. I have read and appreciate the responses made by the authors to my comments and concerns. I have also read the insightful comments and concerns of Reviewer #1 from the prior submission and I am generally satisfied that the authors have addressed those points. The thoughtful discussion contributions made by both parties are very much appreciated.

It is my view that the text of the paper alone reads well at present. It is quite straightforward and I believe that the current version is better than the last.

We thank the reviewer for these comments. It was our hope that the text would read well.

However, I am frustrated about the way the Figures and Tables (including those that are Supplementary) are integrated with one another and the text. In order to follow along, one must open the Results section of the text and open the corresponding (in-text) Figures/Tables and also have the Supplementary Figures/Tables open as well as the Experimental Procedures section of the text to gather all of the information needed to understand a given result. This frustration has been mentioned by other, prior Reviewers (see discussions pertinent to readability and sequence of data presentation). This frustration seems to extend beyond the routine and expected awkwardness of navigating a manuscript before it has been accepted and transformed into the final publication format. It makes me wonder if a full re-working of the manuscript is necessary rather than merely updating it piecemeal to satisfy the comments garnered in each additional round of review. No doubt the document has been reviewed and updated many times.

A key concern I have is that the authors seem to refer to elements of figures (particularly Supplementary Figures), which I cannot find. This makes me wonder if I am looking at the same version of the Supplementary Figures as the authors are looking at. Please see below for line-by-line examples.

I thus conclude that from a scientific standpoint, I am satisfied that my concerns have been addressed. However, I’d encourage the authors to sit with the paper and take a very careful look at all of the references to in-text / Supplementary Figures and Tables to make sure that everything is coherent.

We appreciate the reviewer’s comment about careful checking of the figures and tables, especially in the light of the well-made line by line comments. We have given the manuscript thorough checking to make sure that each reference to figures, tables and supplementary material is correct.

Line-by-line Comments

Lines 95-99: “Previous studies have shown that approximately 0.1% of B. thailandensis cells survive for 24 hours following treatment with 100X MIC.” Likewise, can you quote a survival percentage for B.P. from the literature – if available?

These data are available in the literature, and we have added this to this section (revision – line 101).

Line 166-169: “pIC50, hillslope and maximal effect were used to further down select to six compounds (A-F), all of which displayed a pIC50 > 5 (Figure 3A, Table 1, Supplementary Figure 8, Supplemental Table 1)”. Firstly, it is tremendously confusing to make a statement and then provide substantiation of the statement by referring to an in-text figure, an in-text table, a supplementary figure and a supplementary table. It is almost enough to cause the reader to give up on trying to independently review points of evidence for the claim. There must be a better way to convey which parts of the claim are substantiated by evidence in which in-text figures and tables and in which Supplementary figures and tables. Secondly, I don’t see the structures for Compounds A-F in Supplementary Table 1. Are they supposed to be here or elsewhere – for example in one of the other tables or figures? Moreover, which letter (A-F) is chloroxine in each figure/table?

We appreciate the reviewer’s point about the challenge of viewing these data. To address this, we have broken these data into two, highlighting the primary assay for Fig 3 and Table 1, and the secondary assay in the Supplementary figure/table. [Revision – line 293.]

We have added the structures of compounds A-F to Table S1 as the reviewer suggested.

Chloroxine is compound A – we have clarified this in the text at the point that the reviewer noted, which is a good idea. [Revision – line 294-5.]

Line 179-180: The text suggests that there is a depiction of chloroxine’s antimicrobial activity with an MIC of 4 ug/mL in Supplementary Table 2. I do not see any information regarding this in Supplementary Table 2. Yet again, there is discussion of Reviewer #1’s comments included in the information for the authors from the 2nd submission of this manuscript to PLOS ONE. Here, it sounds like the authors are directing Reviewer #1 to see Supplemental Table 1. I don’t see this information there either. Please address.

We apologise for the lack of clarity here. The CLSI method calls for visual inspection of wells, which does not lend itself well to representation except as a determined MIC. The determined values are listed in Table S1. We have removed the reference to table S2 which was in error. The results shown in Fig S4 are consistent with the determined MIC, providing a further validation of the MIC result.

Figure 1: Please state what the positive control is in the figure legend. Otherwise one must dig and look in the Experimental Procedures section to find it.

This is an excellent suggestion and we have added this to the figure legend. [Revision: lines 251-2.]

Response to Reviewer#2 Comment from Prior Submission: Previously, I requested an example of how the ‘Z’ factor is calculated. In your response, you indicated that you have included one and I was eager to take a look at that. However, I cannot find an example in the legend of Supplementary Figure 1, which is where I was directed to look. This makes me question if you have perhaps submitted the wrong version of the Supplementary Figures and that I am consequently now looking at the wrong version (?). Perhaps I am not looking at the final draft of your updates to the Supplementary Figures (?). This could account for my ongoing confusion. For context, this is what I see:

Supplementary Figure 1: Assay development using Bactiter Glo to measure ATP levels.

ATP levels in an untreated culture of B. thailandensis was quantified with Bactiter Glo reagent, in a series of two-fold dilutions with media. Signal from the Bactiter Glo was converted to an ATP concentration using an ATP standard curve in the same media. Results are the mean of three replicates. Error indicates 95% confidence intervals. Z’ for 0.8 to 0.4 = 0.61.

Thank you for your careful consideration to these matters.

We apologise here, as we unfortunately misunderstood the reviewer’s previous request. We have included a sample calculation in Fig S1, which is the ideal place for a detailed example.

Reviewer #2: This reviewer acknowledges that the author and colleagues have addressed all the comments and concerns posed in the review, included pertinent data, added missing information and corrected typographical errors.

We thank the reviewer for their comments.

---

## [Decision Letter · Decision Letter 1]

22 Feb 2021

Drug screening to identify compounds to act as co-therapies for the treatment of Burkholderia species

PONE-D-20-28523R1

Dear Dr. Harmer,

We’re pleased to inform you that your manuscript has been judged scientifically suitable for publication and will be formally accepted for publication once it meets all outstanding technical requirements.

Kind regards,

R. Mark Wooten, Ph.D.

Academic Editor

PLOS ONE

Additional Editor Comments (optional):

Reviewers' comments:

Reviewer's Responses to Questions

**Comments to the Author**

1. If the authors have adequately addressed your comments raised in a previous round of review and you feel that this manuscript is now acceptable for publication, you may indicate that here to bypass the “Comments to the Author” section, enter your conflict of interest statement in the “Confidential to Editor” section, and submit your "Accept" recommendation.

Reviewer #1: All comments have been addressed

2. Is the manuscript technically sound, and do the data support the conclusions?

Reviewer #1: Yes

3. Has the statistical analysis been performed appropriately and rigorously? 

Reviewer #1: I Don't Know

4. Have the authors made all data underlying the findings in their manuscript fully available?

Reviewer #1: Yes

5. Is the manuscript presented in an intelligible fashion and written in standard English?

Reviewer #1: Yes

6. Review Comments to the Author

Reviewer #1: (No Response)

7. PLOS authors have the option to publish the peer review history of their article (what does this mean?). If published, this will include your full peer review and any attached files.

Reviewer #1: No

---

## [Editor Report · Acceptance letter]

17 Mar 2021

PONE-D-20-28523R1 

Drug screening to identify compounds to act as co-therapies for the treatment of *Burkholderia* species 

Dear Dr. Harmer:

I'm pleased to inform you that your manuscript has been deemed suitable for publication in PLOS ONE. Congratulations! Your manuscript is now with our production department. 

Kind regards, 

on behalf of

Dr. R. Mark Wooten 

Academic Editor

PLOS ONE